# Triadic communication with teenagers and young adults with cancer: a systematic literature review – 'make me feel like I'm not the third person'

Deborah J Critoph [1] , Rachel M Taylor [2] Anna Spathis,[1] Robbie Duschinsky,[1] Helen Hatcher,[3] Ella Clyne,[4] Isla Kuhn,[5] Luke A M Smith [3]

¹Department of Public Health & Primary Care, University of Cambridge, Cambridge, UK
²CNMAR, University College London Hospitals NHS Foundation Trust, London, UK
³Cambridge University Hospitals NHS Foundation Trust, Cambridge, UK
⁴University of Cambridge, Cambridge, UK
⁵Medical Library, School of Clinical Medicine, University of Cambridge, Cambridge, UK

**Correspondence to**
Deborah J Critoph;
dc625@cam.ac.uk

## ABSTRACT

**Objectives** Clinical communication needs of teenagers and young adults with cancer (TYACs) are increasingly recognised to differ significantly from younger children and older adults. We sought to understand who is present with TYACs, TYACs' experiences of triadic communication and its impact. We generated three research questions to focus this review: (1) Who is present with TYACs in healthcare consultations/communication?, (2) What are TYACs' experiences of communication with the supporter present? and (3) What is the impact of a TYAC's supporter being present in the communication?

**Design** Systematic review with narrative synthesis.

**Data sources** The search was conducted across six databases: Medline, CINAHL, Embase, PsycINFO, Web of Science and AMED for all publications up to December 2023.

**Eligibility criteria for selecting studies** Included papers were empirical research published after 2005; participants had malignant disease, diagnosed aged 13–24 years (for over 50% of participants); the research addressed any area of clinical communication.

**Data extraction and synthesis** Three independent reviewers undertook full-text screening. A review-specific data extraction form was used to record participant characteristics and methods from each included paper and results relevant to the three review questions.

**Results** A total of 8480 studies were identified in the search, of which 36 fulfilled the inclusion criteria. We found that mothers were the most common supporter present in clinical communication encounters. TYACs' experiences of triadic communication are paradoxical in nature—the supporter can help or hinder the involvement of the young person in care-related communication. Overall, young people are not included in clinical communication and decisions at their preferred level.

**Conclusion** Triadic communication in TYACs' care is common, complex and dynamic. Due to the degree of challenge and nuances raised, healthcare professionals need further training on effective triadic communication.

**PROSPERO registration number** CRD42022374528.

## STRENGTHS AND LIMITATIONS OF THIS STUDY

⇒ We searched systematically and thoroughly for eligible studies, but this is not a well-indexed field of research, and therefore, it is possible that some relevant studies were not included in the review.

⇒ We limited the review to UK teenagers and young adults with cancer (TYACs) age range and not the broader age used elsewhere, so the conclusions are applicable to younger adults, up to age 24 only and not necessarily the age of young adulthood used in some countries (between 29 and 39).

⇒ We only included papers published in English and the results may not be applicable to other countries especially where cultural differences affect parental–TYAC or other familial/romantic relational dynamics.

⇒ International representation was seen in the eligible studies and TYAC ages were included across the entirety of the specified UK age range.

⇒ Studies represented the journey throughout the cancer experience from diagnosis to survivorship and end-of-life care.

cognitive, emotional and behavioural milestones to develop a sense of self-identity and gain independence. Although most young people have limited encounters with healthcare, around 2500 young people in the UK are diagnosed with cancer each year, which is the leading cause of non-traumatic death in young people in the USA and Europe.[1] Teenagers and young adults with cancer (TYACs) have unique healthcare needs and there has been an international drive to develop developmentally appropriate evidence-based specialist care, provided by appropriately trained healthcare professionals (HCPs).[2]

Communication with TYACs can be particularly challenging: a life-limiting condition intersects an age associated with emotional reactivity and variable maturity. TYACs' clinical communication needs are increasingly

## INTRODUCTION

Adolescence is a time of transition where young people navigate monumental physical,

recognised to differ significantly from younger children and older adults. Research indicates TYACs can have little meaningful involvement in conversations with HCPs: almost half of children and young people reported not being involved in decisions about their care.[3] HCPs recognise this and consider young people among the hardest patients to communicate with.[4] However, HCPs receive little training about how best to manage these clinical encounters. TYACs perceive that HCPs do not make efforts to understand how their cancer impacts their life outside of the healthcare setting. As a result, they may withdraw and subsequently be labelled as 'challenging', 'hard to reach' and 'disengaged'. This may adversely impact care and contribute to poor physical and psychological outcomes. Despite these issues, there are limited opportunities for formal postgraduate education in communication with TYACs for HCPs, with most training being ad hoc and not interprofessional.[5 6] Effective communication with TYACs has been recognised as a key national research priority. In a UK-wide survey of young patients' own research priorities, communication was a striking cross-cutting theme.[7]

Recent research into clinical communication with TYACs has offered some insights into the complexities of communication with this specialist patient group.[8–12] Yet one area that has received less attention is triadic communication. Triadic communication refers to the presence of a third party, such as a parent, carer or companion in clinical encounters[13] and the presence of such a person was found to occur in 87% of TYACs' consultations.[11] As a commonly occurring form of communication in the care of TYACs, there is a need to understand the theoretical basis and relevance of triadic communication to clinical practice. For the purposes of this review, we refer to this third person as a supporter. Triadic communication literature from children and older adults exists.[14–17] Notably this includes a meta-analytic review of provider–patient–companion of adults,[18] one large systematic review of physician–patient–companion communication and decision-making in adults[19] and one review of doctor–parent–child communication.[20] While informative, these studies are with children and adults, not this unique age group of emerging adulthood with a significant life-threatening diagnosis such as cancer. Also, these studies focus on doctor–patient–third person communication, whereas TYAC care involves a range of interdisciplinary professionals. This review aims to understand what is known about triadic communication with TYACs in healthcare communication.

### Aim

We sought to understand who is present with TYACs, synthesise TYACs' experiences of triadic communication with HCPs and supporter(s), and develop insights into the impact of triadic communication for TYACs.

### Review questions

1. Who is the supporter present with TYACs in healthcare consultations and communication?

> **Box 1  Search terms**
>
> Strand 1—TYAC
> TYA cancer or TYA oncology or teenage and young adult adj5 cancer or teenage and young adult adj5 oncology or teenage* adj5 cancer or teenage* adj5 oncology or adolescen* adj 5 cancer or adolescen* adj 5 oncology or young people adj 5 cancer or young people adj 5 oncology
> Strand 2—communication
> Communication skills OR communicat* OR discuss* OR disclos* OR inform* OR interact OR relationship building OR decision making OR communication tools OR communication aids OR psychosocial assessment
> Strand 3—supporters
> Parent* or guardian* or mother* or father* or partner or wife or wives or husband* or boyfriend* or girlfriend* or sibling* or friend* or carer* or "third person" or caregiver* or "care-giver*" or spouse* or supporter* or support network*.
> Strand 4—impact
> affect OR effect OR influence OR result OR resultant OR impact
> Strand 5—experience
> encounter OR involvement OR occurrence OR feel OR "go through" OR experience*
>
> TYAC, teenagers and young adults with cancer.

2. What are TYACs' experiences of communication with the supporter present?
3. What is the impact on a TYAC's supporter being present in the communication?

### METHODS

We conducted a systematic review and narrative synthesis[21 22] of empirical evidence published since 2005, the year of publication of the National Institute for Care Excellence Improving Outcomes Guidance, the guidance document underpinning TYAC services in England.[2] The review protocol was prospectively registered with PROSPERO. We designed the search to identify and map the available evidence using a broad scope to gain an overview of the pertinent literature, identify knowledge gaps and clarify concepts. The search strategy was developed and refined with an information scientist (IK). Keywords were generated across five strands detailed in box 1, with strands combined with the Boolean operator 'AND'. The search was conducted across six databases: Medline, CINAHL, Embase, PsycINFO, Web of Science and AMED (online supplemental file 1).

Database searches were compiled and de-duplicated in Mendeley, abstracts were screened in Rayyan by two researchers (DJC and LAMS), and 172 full articles were read by three researchers (LAMS, DJC and RMT) for eligibility of inclusion in the final analysis, with disagreements resolved by discussion. Papers were included if: they presented empirical research published after 2005; participants had malignant disease, diagnosed aged 13–24 years (for over 50% of participants); the research addressed any area of clinical communication and the research included supporters (parents, partners, carers, friends, etc). Papers were excluded if they were:

conference abstracts, unpublished articles, systematic reviews, single case studies, validation research methodology, studies using retrospective documentation in clinical notes, articles focusing on information needs rather than communication skills or were not in English.

A review-specific data extraction form was used to record participant characteristics and methods from each included paper and results relevant to the three review questions. The final number of included articles totalled 36, the remaining 136 were excluded based on the participants' ages, focus on HCPs or information giving. In tandem to the data extraction process, two members of the review team (EC and DJC) independently assessed each paper in terms of its internal validity, appropriateness and contribution to answering the review questions, using a review-specific version of Gough's Weight of Evidence criteria.[23] Discrepancies in assessment decisions were discussed between reviewers and final scores were agreed through consensus.

Extracted data were entered into Excel to aid the narrative synthesis of the included papers.[21 22] All articles, irrespective of relevance and quality, were included in the review. However, those rated 'medium' and 'high' were given greater weight in the synthesis. An inductive thematic analysis was undertaken to identify the main, recurrent and important data across the studies related to answering each research question. DJC and EC explored heterogeneity across the studies. The integration of results from studies using different methods and epistemological positions was supported by LAMS and RMT, and consensus in synthesis was reached. The synthesis was further refined through discussion of the review of results and their implications with clinicians, interdisciplinary academic audiences and all of the co-authors.

### Patient and public involvement statement
None.

### RESULTS
A total of 8480 studies were identified in the search, of which 36 fulfilled the inclusion criteria (figure 1). The included articles are summarised in online supplemental table 2.

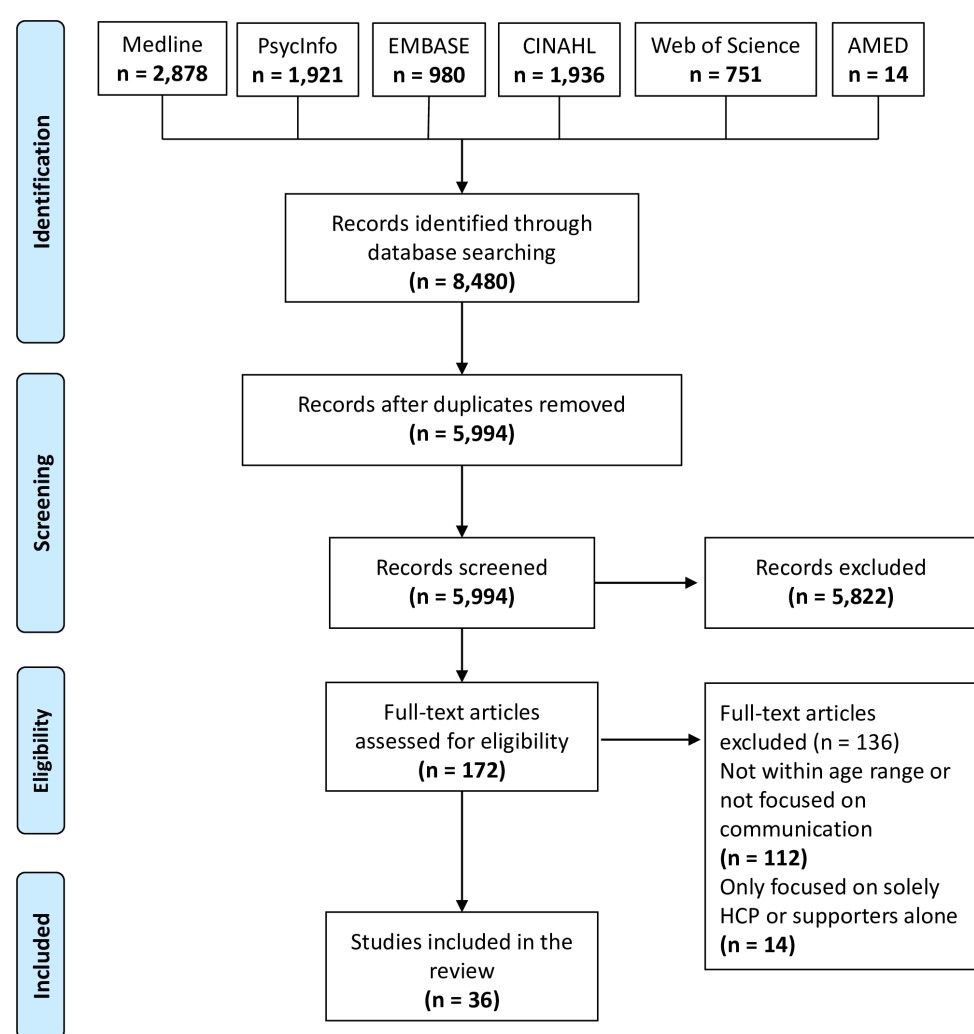

**Figure 1** PRISMA flow diagram. HCP, healthcare professional; PRISMA, Preferred Reporting Items for Systematic Reviews and Meta-Analyses.

All points across the cancer trajectory were represented in the final papers: diagnosis (n=7)[12 24–29]; on treatment (n=17)[30–46]; end of treatment (completed within 1 year) (n=2)[47 48]; survivorship (more than 1 year post-treatment) (n=2)[5 49] and end-of-life care (n=5).[50–54] Three studies included patients at more than one point along the cancer care continuum.[55–57] Most studies (n=19) were conducted in the USA[24 27–29 31 35–37 39–46 50 52 54] other countries included the UK,[25 32 33] Australia,[38 48 49 57] Norway,[12 53] Israel,[47] Iran,[30] Mexico,[51] France,[34] Denmark,[26] Korea[56] and Taiwan,[55] one study recruited from three European countries.[5] Studies used predominantly qualitative methods (n=32) but there were two mixed-methods studies and two using quantitative methods. Weight of evidence (WoE) criteria indicated 5 were high evidence,[24 31 35 45 56] 24 were medium[5 12 25 27–30 32–34 36 37 39–42 44 46 47 49–51 55 57] and 7 were low evidence.[26 38 43 48 52–54] We used Gough's review-specific criteria to weigh the quality of each paper.[23] To do this, we used three parameters:

1. The integrity of the evidence on its own terms.
2. The appropriateness of the method for answering the review questions.
3. The appropriateness of the focus or relevance for answering the review questions.

Each of the above was either rated as low, medium or high. These three parameters were combined to create WoE D which was the overall rating seen above and is the extent to which a study contributes evidence to answering the review questions. Factors that made the method highly appropriate included the use of semistructured interviews to understand TYACs' experiences and speaking to the TYAC and supporter separately. The high-scoring papers included papers that focused on communication in the triad, but this only occurred in 10 papers. In nine papers, the age at diagnosis was not specified and this decreased the weighting of these papers.[5 34–36 50–54]

Of the included studies, just under one-third researched the triad (n=10) of TYACs, supporters and HCPs,[5 24 30–32 34–36 50 51] one-third TYACs only (n=12)[28 29 33 37 38 40–42 44–46 48] and just over one-third TYACs and supporters (n=14)[12 25–27 39 43 47 49 52–57] (see table 1).

The categories used to separate the age groups were lower adolescence (11–14 years), middle adolescence (15–17 years), upper adolescence (18–21 years) and emerging adulthood (22 onwards). Of the papers where the age range at diagnosis could be deduced, the majority of these (21 out of 24) spanned three or more age categories. All the papers spanned two or more age categories. In nine of the papers, the age ranges at diagnosis were not available (as age at diagnosis was expressed as a mean or median). Given these factors, it is difficult to ascertain whether any between age group differences exist.

## Who is present with TYACs in healthcare consultations and communication?

The majority of supporters were mothers (68.9%). When combined, parents represented nearly all the supporters in the included studies (94.6%), see table 2. Non-parental supporters (1.8%) included partners, sisters, aunts and grandmothers. The remaining supporters were not categorised due to insufficient information in the article's demographics data (3.9%).[53 54]

## What are TYACs' experiences of communication with the supporter present?

The presence of supporters was concurrently helpful and challenging for TYACs. Supporters undertook several helpful roles and responsibilities: they asked questions on behalf of the TYACs, retained information from HCPs, acted as a conduit of information between the TYACs and HCPs, and acted as a 'sounding board' for the young person.[25 31 45] Some supporters promoted self-advocacy and autonomy for the young person.[27 39 41 46 57] Some reported symptoms on their behalf[45] and proactively negotiated changes to treatment schedules in the interest of the young person.[39]

Findings also suggested that young people could experience limited or ineffective communication in the presence of a supporter. Communication could be directed towards the supporter, not the young person.[27 29 31 36] Supporters could receive information in the absence of the TYACs and subsequently filter the content before delivering the information to TYACs.[30 33 34 55 56]: 'The parents had hidden a truth that was not theirs to hide' (p 533).[34] This reflected the broader predicament that supporters' priorities at times might have competed with those of young people.[25 34 50 51] Supporters could dominate the communication encounter, for instance, parents were seen to interrupt young people, especially when time was limited.[51] Frederick *et al* found the mean time for adolescent to clinician communication was only 5.5% of the total consultation and parent conversation turns directed towards clinicians comprised a mean of

**Table 1** Study population

| Triad? Dyad? Single? | Who is studied in the paper? | Number of papers | References |
|---|---|---|---|
| Triad | TYACs, supporter, HCPs | 10 | 5 24 30–32 34–36 50 51 |
| Dyad | TYACs and supporter | 14 | 12 25–27 39 43 47 49 52–57 |
| Single | TYACs only | 12 | 28 29 33 37 38 40–42 44–46 48 |

Participants included in the study and the number of papers included for each of the three participant groups.
HCPs, healthcare professionals; TYACs, teenagers and young adults with cancer.

**Table 2** Supporter demographics

| Supporter type | Number of supporters | Percentage quoted to one decimal place (%) | References |
|---|---|---|---|
| Mother | 453 | 68.9 | 5 12 24–27 30–32 34 36 39 43 47 49–52 54 55 |
| Father | 128 | 19.5 | 5 12 25–27 30–32 34 36 39 43 47 49–52 55–57 |
| Both parents | 20 | 3.0 | 12 32 34 36 55 |
| Parents—no further specification | 20 | 3.0 | 35 |
| Stepmother | 1 | 0.2 | 57 |
| Grandmother | 2 | 0.3 | 24 |
| Sister | 3 | 0.5 | 12 30 51 |
| Partner | 3 | 0.5 | 25 52 |
| Aunt | 3 | 0.5 | 36 51 52 |
| Supporters—no further specification | 21 | 3.2 | 53 54 |
| Other | 3 | 0.5 | 55 |
| Total | 657 | 100.1 | |

Details of the supporter demographics and percentages within the included publications.

37.5% of all conversation turns. Clinicians directed most communication at the parent rather than the adolescent and spoke for 66.9% of the conversation and none of the clinicians offered patients the opportunity to speak with them alone.[35]

Mutual protectionism appeared to occur, with TYACs and supporters seeking to protect each other from difficult information leading to non-disclosure when both were present. A diagnosis of cancer is devastating for the young person, supporter(s), family and the wider social network. Repeatedly, there were references to reduced disclosure between the young person and their supporter, in an attempt to shield each other from emotional distress.[12 31 36 38 39 41 45 53 56] TYACs could experience discomfort and guilt in seeing parents tearful and worried, and felt a burden in response to observing the emotions of supporters.[38 39 52] Some TYACs sought to limit this by withholding concerns to protect their supporters: 'I couldn't talk to mum about my concerns because I didn't want to hurt her' (p 37).[38] In equal measure, supporters were characterised as working hard to stay in control of emotions, be strong and stay in the 'now', and they channelled energy into helping.[12 31 56] Yet, this could contribute to an environment of non-disclosure that had the potential to create future communication challenges, such as supporters not knowing the young person's wishes. Examples of this were evident within the end-of-life care studies.[52 53] Friebert *et al* found that 86% of young people wanted to receive prognostic information as soon as possible but only 39% of families knew that.[52] Similarly, Jacobs *et al* found that young people's end-of-life wishes were not known by their families.[53] In instances where the young person may not be able to communicate, it may help families relieve the impossible burden of making difficult decisions or feelings of regret, if the young person's perspective and wishes are known.[54]

## What is the impact of a TYAC's supporter being present in the communication?

Supporters have the potential to facilitate, complicate or obstruct the young person's involvement in decision-making. Involvement had a positive impact on recall,[42] and may improve autonomy, efficacy, adherence and future self-management.[24 57] However, the participation of supporters may be experienced as stressful by TYACs as they may become side-lined.[25 40 55] The presence of supporters impacted the young person's level of involvement in decision-making in several ways. In some cases, supporters empowered TYACs to make decisions by withholding their opinion[27] and deferring the final decision to TYACs.[31] However, supporters and TYACs did not perceive decision-making in the same way.[47 56] Supporters believed that young people oversaw decisions about their care; however, this was not what young people recounted.[24] TYACs reported a lack of communication and limited involvement in decisions[24 29 30 46] associated later with decisional regret.[24 37]

Deferral of communication and decisions from the young person to supporters was commonplace.[27 31 36] When supporters responded to this pathway of communication, young people then did not see a need to participate in decisions, knowing that their supporter was taking the mantle.[36] In parallel, clinicians were found to direct communication towards supporters and in extreme cases, young people were completely excluded from communication and decisions.[29 30 35 47] An atmosphere characterised by a lack of trust, unanswered questions and uncertainty contributed to the exclusion of young people who then sought information from other sources.[30 36 39 56] Not allowing TYACs to choose their involvement in decision-making violated their autonomy, and increased distrust or resentment of providers and supporters and resulted in lower treatment adherence.[30 36 39]

The decisional involvement preferences of young people were not static: they were context and environment dependent. At diagnosis, heightened emotions and poor health rendered young people unable to engage in communication.[24 25 27 29 31 37 41] TYACs expressed a desire to be involved in decision-making at different levels: some wanted limited involvement from their supporter(s) so they could take the leading role in consultations and their care[37]; several wanted collaboration with supporters and clinicians[26 27 44 57] and some completely relied on supporters and HCPs to make decisions on their behalf.[45 46] Davies *et al* described this as agency, the ability to make free and independent choices. They highlighted the normality of this fluctuation between personal (acting independently), proxy (decisions made on behalf of someone) and collective (decisions are shared) decision-making. While this was not always linear, it was part of the cancer trajectory and demonstrated the fluctuating personal agency for TYACs.[32] Some young people reported that supporters and clinicians decided on their level of involvement in communication and decision-making,[55] and TYACs commented that they did not feel the decision was theirs.[47] Decisional involvement was an interactive, complex and multifaceted process within the context of the triad, and young people often wanted to be in control of their level of involvement.[28 31] The evidence highlighted that in the presence of a supporter, young people's choice in their level of involvement in decisions was challenged and not routinely achieved.

Most TYACs felt that it was important for the healthcare team to communicate with them directly and openly.[30 31 33 38 39 49 50] Time alone helped facilitate communication between TYACs and HCPs, to ensure that the young person's needs were fully met.[31 36] However, time alone with HCPs was not routinely integrated as a part of consultations with TYACs.[35 48] In fact, clinicians were reported as frequently speaking more to parents and TYACs received limited communication from HCPs.[27 31 35 36] In the presence of supporters, as well as withholding concerning information, young people reported feeling discomfort when discussing sensitive topics such as sex or fertility preservation.[27 36]

Young people wanted time alone to communicate with HCPs directly for a variety of reasons. This private line of communication offered a sense of personal agency and allowed them to feel 'in the loop' and promoted a sense of autonomy that was threatened by the cancer diagnosis, particularly at the point of diagnosis.[32 50] Young people wanted space to think and privacy during the cancer journey; private lines of communication with HCPs actively promoted this.[31 39 45 46] It also enabled HCPs to get to know the young person and allowed them to ask questions that they may be reluctant to ask in the presence of their supporter, because of embarrassment or emotional shielding.[31] Darabos *et al* found that 87.5% of oncology providers considered it important to talk to the TYACs without their parents present.[31] While the importance has been highlighted within the data it is also evident that this does not happen as part of routine clinical practice. This could be for several reasons such as not wanting to challenge rules of authority, uncertainty around how best to ask a parent to leave and lack of confidence when communicating with a young person alone.

## DISCUSSION

### Principal findings

#### Who is present with TYACs in healthcare consultations and communication? For example, who are the supporters?

The included papers in our review demonstrated that most supporters were parents, more commonly mothers. The frequent presence of mothers in consultations is consistent with previous findings. For example, in a UK study in which TYAC nominated a caregiver, 85% were parents, and of those 80% were women.[58] We note that there is a paucity of data for non-parental supporters, and this may represent a reality of clinical practice or a bias towards TYAC–parental dyads over other relational dyads in this field of research to date.

#### What are TYACs' experiences of communication with the supporter present?

TYACs experienced supporters facilitating communication by obtaining information, asking questions, advocating and supporting personal agency of the young person; conversely, supporters could hinder communication by gatekeeping information, or dominating communication and thereby rendering young people as bystanders. Young people experienced negative emotions in response to witnessing their supporters in distress.

#### What is the impact of a TYAC's supporter being present in the communication?

Bidirectional non-disclosure was a coping strategy used by both TYACs and supporters to protect one another from concerns and emotional burdens. This limited HCPs' ability to effectively assess ideas, concerns and expectations from both parties when together. In the presence of supporters, some young people were less informed, which could impair their ability to engage in decision-making conversations.

### Meaning of the study

This is the first review to look specifically at triadic communication in TYACs and has demonstrated that there is a paucity of evidence focused specifically on triadic communication with TYACs. Of the 36 studies in the review less than one-third included all three parties in the triadic communication encounter. However, the review has enabled us to provide answers to the review questions and identify knowledge gaps, including a lack of theory describing triadic communication. Some preliminary theoretical models, such as family involvement in interpersonal healthcare processes,[59] depict the interaction pathways between patients, families and HCPs

and hypothesise the influence of family on interpersonal processes and outcomes of medical consultations.

The data clearly identified that parents are the predominating supporters for TYACs, which may be surprising given the inclusion of participants up to the age of 25. Parents can play a significant role when a young person is diagnosed with cancer. Developmentally, a major characteristic that differentiates TYACs from younger children or older adults is the progressive increase in their desire and capacity for independence, personal agency and autonomy. This process is disrupted by a cancer diagnosis: increased parental presence can be perceived as intrusive and reflect reversion to an earlier family dynamic, anchoring TYACs in dependency, restricting self-exploration and limiting their development of an internal value and belief system.[38 60–62] This has been phrased as 'retreating to family' and can negatively impact peer relationships by impeding the development and maintenance of a peer network.[40 63 64] Young people may often be accepting of this, particularly in the early stages of the cancer diagnosis. However, as this review demonstrates, the presence of parents alters the experience and impact of communication with HCPs. It is important to highlight that there is limited literature on TYAC communication encounters with supporters other than parents.[61 65 66] Partners felt relegated to a non-participatory role by a parent, and mothers struggled to relinquish their existing role as primary supporter.[61 66] It is relevant to note that the participants in these three studies were in their early 20s.

A key impact of triadic communication is that young people may not be involved in decision-making to the level they want. This is consistent with related paediatric oncology literature which consistently reports children's limited participation in decision-making.[67–69] Clinicians attempted to protect children from 'too much' information because of the perception that children are not capable or too vulnerable.[17] The important difference between paediatric and TYAC populations is the legal and ethical obligations towards TYACs who are autonomous, capacitous patients rather than to parents with parental responsibility.

The findings of this review demonstrate the presence of a supporter impacts the involvement of young people in healthcare decisions. Therefore, there are legal and ethical issues, which are critically important, both in research and clinically in TYAC care particularly related to informed consent, capacity, and autonomy. The law relating to children and young people is complex and differs across the UK and internationally. The General Medical Council guidelines in the UK state, 'the patient must be the first concern'.[70] HCPs have ethical and legal obligations outlined in the UK best practice guidance, statute and case law.[71] In the UK, parents can legally make decisions for children under 16 years unless the child disagrees and is deemed 'Gillick Competent'.[72] Moreover, studies have shown children aged 14 and older can approach the level of understanding of adults.[73 74]

In contrast, people aged 16 and above are legally able to make decisions for themselves in the UK and are automatically assumed to have capacity[75] and therefore, HCPs must communicate with them in developmentally appropriate ways. Clinicians face a challenge in identifying the best way to communicate with TYACs and their supporter(s). TYACs need parental involvement while simultaneously desiring autonomy[36] necessitating careful balancing of the needs of both parties to ensure that the young person is not relegated to a non-participant status.

## Strengths and weaknesses

Our review had several limitations. We searched systematically and thoroughly for eligible studies, but this is not a well-indexed field of research, and therefore it is possible that some relevant studies were not included in the review. We limited the review to a UK TYAC age range and not the broader age used elsewhere, so the conclusions are applicable to younger adults, up to age 24 only and not necessarily the age of young adulthood used in some countries (between 29 and 39). We also only included papers published in English and therefore papers reflect practices in primarily North America, Australia and Europe, the results may not be applicable to other countries especially where cultural differences affect parental–TYAC or other familial/romantic relational dynamics and where the healthcare culture may be different, for example, more paternalistic. Despite these limitations, international representation was seen in the eligible studies, TYAC ages were included across the entirety of the specified UK age range and studies represented the journey throughout the cancer experience.

## Implications for clinicians and policymakers

Given the degree of challenge and nuance raised, HCPs need training on effective triadic communication. Fourneret concluded that the relationship between TYACs, their parents and HCPs 'as being the most difficult one in oncology'.[34] Professionals described challenges communicating with both TYACs and parents, especially when loyalties were torn between the two.[5] However, training is currently ad hoc and not interdisciplinary.[5 76–78] Furthermore, HCPs can find it difficult to apply teaching in this area in clinical practice.[53 79] HCPs need education and training to navigate triadic communication to optimise the involvement of the young person while attending to a supporter's needs. Experiential learning is the gold standard in teaching methods for clinical communication and is designed to bring about changes in learners' skills. These evidence-based methods are through small group, problem-based simulation in a classroom, with repeated practice and rehearsal of skills under observation with detailed and descriptive feedback. This is arguably warranted here.[80 81]

Triadic communication is a key feature of TYAC care but requires further attention and inclusion in future iterations of key policy documents and guidelines such as the Blueprint of Care (BoC).[82] The BoC is a UK document

that helps shape and deliver developmentally appropriate care to TYACs. However, it is recognised that age is poorly correlated with developmental maturity and therefore any communication framework needs to be specific to TYACs, recognising the transitional nature of adolescence meaning a one-size-fits-all approach is likely inadequate.

## Unanswered questions and future research

Future research is warranted to triangulate triadic perspectives and understand more about the inter-actional dynamics of these complex communication encounters. A key research need is investigating how best to support decision-making while engaging supporters, understanding their priorities and information needs may conflict.[31 36 37 40] Conflict management must also be understood in the emotional context of young adult oncology. How to effectively educate HCPs to communicate within the triad, to ensure the young person and the supporter's needs are met is a priority. This needs to include how best we facilitate time alone between young patients and HCPs. Continued development and utilisation of comprehensive triadic theoretical frameworks may provide guidance and direction for future research, allowing for greater integration and progress with this diverse research area and commonly occurring form of healthcare communication.

## CONCLUSION

Triadic communication is a pivotal component of communicating with TYACs and the presence of supporters impacts clinical communication both positively and negatively. Young people desire a sense of personal agency, autonomy and control related to information flow and decision-making. This includes private lines of communication with HCPs without the presence of supporters. HCPs recognise the importance of time alone with young people; however, this does not translate to clinical practice. Therefore, further research on communication dynamics is needed to allow for the development of bespoke, TYAC-focused clinical communication training for HCPs to allow them to effectively facilitate and navigate triadic communication. This then needs to be formally embedded in national guidance and postgraduate training for HCPs working in TYAC care to allow equitable access for TYACs.

**Contributors** All authors meet the criteria for authorship. DJC, LAMS and RMT were involved in developing the protocol. DJC, IK and LAMS coordinated the running of the study and were responsible for data acquisition. DJC, LAMS, RMT and EC contributed to the analysis. DJC drafted the manuscript. All authors have critically reviewed the manuscript for important intellectual content and have read and agreed to the published version of the manuscript. All authors agree to be accountable for all aspects of the work in ensuring that questions related to the accuracy or integrity of any part of the work are appropriately investigated and resolved. DJC is responsible for the overall content as guarantor.

**Funding** This paper presents work supported by the Wellcome Trust (grant number G115288) under its Programme PhD for healthcare professionals course awarded to the first author DJC, University of Cambridge. RMT is partially funded through UCLH Charity. The views expressed are those of the author(s) and not necessarily those of the Wellcome Trust or UCLH Charity.

**Competing interests** None declared.

**Patient and public involvement** Patients and/or the public were not involved in the design, or conduct, or reporting, or dissemination plans of this research.

**Patient consent for publication** Not applicable.

**Ethics approval** Not applicable.

**Provenance and peer review** Not commissioned; externally peer reviewed.

**Data availability statement** All data relevant to the study are included in the article or uploaded as supplementary information. No previously unpublished primary data are included in the paper. All data relevant to the systematic review are included in the paper or uploaded as supplementary information.

**ORCID iDs**
Deborah J Critoph http://orcid.org/0000-0003-3434-1762
Rachel M Taylor http://orcid.org/0000-0002-0853-0925
Luke A M Smith http://orcid.org/0000-0002-7577-2764

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
