## [Reviewer comments · BMJ Open]

ARTICLE DETAILS

TITLE (PROVISIONAL)	Triadic communication with teenagers and young adults with cancer: a systematic literature review: "Make me feel like I'm not the third person"
AUTHORS	Critoph, Deborah; Taylor, Rachel; Spathis, Anna; Duschinsky, Robbie; Hatcher, Helen; Clyne, Ella; Kuhn, Isla; Smith, Luke

VERSION 1 – REVIEW

REVIEWER	Hanks, Christopher Ohio State University, Internal Medicine and Pediatrics
REVIEW RETURNED	27-Oct-2023

GENERAL COMMENTS	Overall I think this is a well written manuscript covering an important topic about which not enough is known. This manuscript provides an apparently comprehensive summary and identifies opportunities for practice improvement and further study. Most of it was easy to read and understand. There were a few small areas that need clarification. These are listed below: Manuscript Page 3 line 51-52 – The sentence "TYACs who are not heard or understood can be labelled as 'challenging', 'hard to reach' and 'disengaged' was confusing to me. Please clarify what was meant by "not heard or understood". You discuss using a review-specific version of Gough's Weight of Evidence criteria to assign a weight to included articles and thus give those rated medium or high greater weight in synthesis. However, with the information provided in this manuscript and supplemental materials, the authors do not provide transparency into what factors they used to define an article's weight. Thus, the reader is left with no way of assessing whether they agree with the weighting or replicating this work. Further explanation of the weighting process and how that was applied to the synthesis process is needed. I do appreciate that in the results section, the references of different weights are listed. However, it would be helpful to include a column for weighting in Table 2 where each article is summarized. In the results section, the authors note that 37.5% of all conversations turns were parent-to-clinician and that 5.5% of total consultation was adolescent to clinician based on one include article. What was the remaining percentage of the consultation? I realize this is a summary of one of the included articles and that the point of inclusion was to highlight that young people had limited opportunity to communicate directly with the clinician, but
--

	with your current description, we are left wondering what the majority of the conversations turns were comprised of.
--	--

REVIEWER	Fletcher, Chloe University of South Australia, Department of Rural Health
REVIEW RETURNED	23-Nov-2023

GENERAL COMMENTS	Thank you for the opportunity to review this paper, exploring triadic communication with teenagers and young adults with cancer (TYACs) within the healthcare context. The authors conducted a systematic review to explore three research questions: (1) who is present with TYACs in healthcare consultations and communication, (2) what are TYACs experiences of communication with supporters present, and (3) what is the impact of supporters being present on the communication? This is a valuable topic with findings that are relevant for clinical practice. I have some comments below that may help to improve the paper prior to publication.  1. Table 1 uses sub-headings for each strand (e.g., Strand 1 – TYAC), but the key words below don't appear to always match the sub-heading – e.g., Strand 2 – Communication is followed by key words describing cancer and oncology. Is this an error? I might be misinterpreting the table. 2. Personally, I didn't feel that there was any value in including Table 3 – you might consider whether this could be excluded. 3. It might be useful to include a fourth column in Table 4 with citations for studies where each supporter type was included. 4. Your Results include discussion of what TYACs wanted in communication with their HCPs – is this something you intentionally looked for in the reviewed studies? It could be worth signposting this separately from the findings on the impact of supporters being presenting during communication with HCPs. 5. You mention in the Discussion that only one third of the reviewed studies included all three parties in the triadic communication encounter – I was surprised to read this. I think it would be worth mentioning in your Results what proportions of the reviewed studies explored communication between which parties – e.g., TYACs, TYACs-Supporters, TYACs-HCPs, TYACs-Supporters-HCPs, etc. A table could facilitate this. 6. You recommend the use of experiential learning as a method for teaching clinical communication – do you have any specific recommendations for what should be included in a training of this type, based on the review findings? For example, direct and open communication with TYACs, time alone with TYACs, and involving TYACs in decision-making. 7. Page 11, lines 6-8: Change "...anchoring adolescents in dependency, restricting self-exploration, and limiting development of a TYACs internal value and belief system" to "...anchoring TYACs in dependency, restricting self-exploration, and limiting their developing of an internal value and belief system". 8. Page 11, lines 9-11: Change "This has been phrased as 'retreating to family' and can impede the maintenance of sustaining a network of peers and cancer negatively impacts peer relationships" to "This has been phrased as 'retreating to family' and can negatively impact peer relationships by impeding development and maintenance of a peer network". 9. Page 11, line 12: Add "...as this review demonstrates..." after 'However'
--

	10. Page 11, line 16: Use of 'supporting role' when referring to partners feeling sidelined might be confusing given that parents/partners/others are all referred to as 'supporters' throughout the paper.
--	---

VERSION 1 – AUTHOR RESPONSE

Reviewer: 1

Dr. Christopher Hanks, Ohio State University

Comments to the Author:

Overall I think this is a well written manuscript covering an important topic about which not enough is known. This manuscript provides an apparently comprehensive summary and identifies opportunities for practice improvement and further study. Most of it was easy to read and understand.

Thank you for your overarching comments and we are delighted that you agree that the review identifies opportunities for improvements in clinical practice and further study.

There were a few small areas that need clarification. These are listed below:

Manuscript Page 3 line 51-52 – The sentence “TYACs who are not heard or understood can be labelled as ‘challenging’, ‘hard to reach’ and ‘disengaged’ was confusing to me. Please clarify what was meant by “not heard or understood”.

Thank you for these comments. This sentence refers to young people who are not listened to and as a result become disengaged because they stop communicating as they do not feel heard and if they are not heard they are not understood. We have amended in the manuscript to ‘TYACs perceive that HCPs do not make efforts to understand how their cancer impacts their life outside of the healthcare setting’.

You discuss using a review-specific version of Gough’s Weight of Evidence criteria to assign a weight to included articles and thus give those rated medium or high greater weight in synthesis. However, with the information provided in this manuscript and supplemental materials, the authors do not provide transparency into what factors they used to define an article’s weight. Thus, the reader is left with no way of assessing whether they agree with the weighting or replicating this work. Further explanation of the weighting process and how that was applied to the synthesis process is needed. I do appreciate that in the results section, the references of different weights are listed. However, it would be helpful to include a column for weighting in Table 2 where each article is summarized.

Thank you for these comments. We have included additional information in the manuscript and added a column to Table 2.

In the results section, the authors note that 37.5% of all conversations turns were parent-to-clinician and that 5.5% of total consultation was adolescent to clinician based on one include article. What was the remaining percentage of the consultation? I realize this is a summary of one of the included articles and that the point of inclusion was to highlight that young people had limited opportunity to communicate directly with the clinician, but with your current description, we are left wondering what the majority of the conversations turns were comprised of.

Thank you for raising this helpful point. We have now included on page 8 that 66.9% of the conversation was clinicians speaking.

Reviewer: 2

Dr. Chloe Fletcher, University of South Australia

Comments to the Author:

Thank you for the opportunity to review this paper, exploring triadic communication with teenagers and young adults with cancer (TYACs) within the healthcare context. The authors conducted a systematic review to explore three research questions: (1) who is present with TYACs in healthcare consultations and communication, (2) what are TYACs experiences of communication with supporters present, and (3) what is the impact of supporters being present on the communication? This is a valuable topic with findings that are relevant for clinical practice. I have some comments below that may help to improve the paper prior to publication.

Thank you for your overarching comments, we are pleased that you agree this is a valuable topic and that the findings are relevant for clinical practice.

1. Table 1 uses sub-headings for each strand (e.g., Strand 1 – TYAC), but the key words below don't appear to always match the sub-heading – e.g., Strand 2 – Communication is followed by key words describing cancer and oncology. Is this an error? I might be misinterpreting the table.

Thank you for drawing our attention to this error and oversight. This has now been amended.

2. Personally, I didn't feel that there was any value in including Table 3 – you might consider whether this could be excluded.

We have now excluded this.

3. It might be useful to include a fourth column in Table 4 with citations for studies where each supporter type was included.

This has now been included thank you for this suggestion.

4. Your Results include discussion of what TYACs wanted in communication with their HCPs – is this something you intentionally looked for in the reviewed studies? It could be worth signposting this separately from the findings on the impact of supporters being presenting during communication with HCPs.

This is part of what young people want when communicating with HCP whether a supporter or is present or not. The results tell us that sometimes this is facilitated and that sometimes this is hindered by the presence of a supporter and that this is inextricably linked to the experiences and impact of a supporter being present, therefore we have not amended the manuscript.

5. You mention in the Discussion that only one third of the reviewed studies included all three parties in the triadic communication encounter – I was surprised to read this. I think it would be worth mentioning in your Results what proportions of the reviewed studies explored communication between which parties – e.g., TYACs, TYACs-Supporters, TYACs-HCPs, TYACs-Supporters-HCPs, etc. A table could facilitate this.

We have now included these in the results section, thank you for drawing to our attention that this was an oversight and not included in the results when this is highly relevant as it was a surprise to us also. We have also included a table.

6. You recommend the use of experiential learning as a method for teaching clinical communication –

do you have any specific recommendations for what should be included in a training of this type, based on the review findings? For example, direct and open communication with TYACs, time alone with TYACs, and involving TYACs in decision-making.

Thank you for this comment. This was raised as the evidence suggests this is the gold standard way of teaching clinical communication skills. We do not yet, have any specific recommendations because the review has highlighted that there is still much, we do not know. For example, how to include young people in decisions and still include supporters, how to navigate conflict, how to facilitate time alone. We have identified on page 12-13 that how and what we train professionals is one of several next research priorities. The lead author's doctoral study aims to address this.

7. Page 11, lines 6-8: Change "...anchoring adolescents in dependency, restricting self-exploration, and limiting development of a TYACs internal value and belief system" to "...anchoring TYACs in dependency, restricting self-exploration, and limiting their developing of an internal value and belief system".

This has been amended, thank you.

8. Page 11, lines 9-11: Change "This has been phrased as 'retreating to family' and can impede the maintenance of sustaining a network of peers and cancer negatively impacts peer relationships" to "This has been phrased as 'retreating to family' and can negatively impact peer relationships by impeding development and maintenance of a peer network".

This has been amended, thank you.

9. Page 11, line 12: Add "...as this review demonstrates..." after 'However'

This has now been included, thank you.

10. Page 11, line 16: Use of 'supporting role' when referring to partners feeling sidelined might be confusing given that parents/partners/others are all referred to as 'supporters' throughout the paper.

Thank you, this has been re-phrased to a non-participatory role.

Reviewer: 1

Competing interests of Reviewer: None

Reviewer: 2

Competing interests of Reviewer: No competing interests

VERSION 2 – REVIEW

REVIEWER	Fletcher, Chloe University of South Australia, Department of Rural Health
REVIEW RETURNED	18-Jan-2024

GENERAL COMMENTS	Thank you to the authors for making the suggested amendments. It is good to hear that this research is part of a broader program addressing training clinical communication skills. I wish the lead author all the best for their doctoral research. I noticed two small things while reviewing this revision:
---

	 - Table 3: Would it be possible for the authors to include a fourth column with the citations of the papers who studied each of the participant groups (triad / dyad / single)? - Table 4: Similarly, would it be possible for the authors to include a fourth column with the citations of the papers who studied each of supporter type? - Discussion: Check sub-heading related to second research question - I think some text have been accidentally deleted.
--	--

VERSION 2 – AUTHOR RESPONSE

Reviewer: 2

Dr. Chloe Fletcher, University of South Australia

Comments to the Author:

Thank you to the authors for making the suggested amendments. It is good to hear that this research is part of a broader program addressing training clinical communication skills. I wish the lead author all the best for their doctoral research.

Thank you very much for your encouragement.

I noticed two small things while reviewing this revision:

- Table 3: Would it be possible for the authors to include a fourth column with the citations of the papers who studied each of the participant groups (triad / dyad / single)?

This has now been included, thank you for this suggestion.

- Table 4: Similarly, would it be possible for the authors to include a fourth column with the citations of the papers who studied each of supporter type?

This has now been included, thank you for this suggestion.

- Discussion: Check sub-heading related to second research question - I think some text have been accidentally deleted.

Thank you for noticing this oversight - this has now been amended.

Reviewer: 2

Competing interests of Reviewer: N/A